# Imaging Microstructural Parameters of Breast Tumor in Patient Using Time-Dependent Diffusion: A Feasibility Study

**DOI:** 10.3390/diagnostics15070823

**Published:** 2025-03-24

**Authors:** Shuyi Peng, Peng Sun, Jie Liu, Juan Tao, Wenying Zhu, Fan Yang

**Affiliations:** 1Department of Radiology, Union Hospital, Tongji Medical College, Huazhong University of Science and Technology, Jiefang Avenue 1277, Wuhan 430022, China; shuyipeng@hust.edu.cn (S.P.); liu_jie0823@163.com (J.L.); 2014xh0917@hust.edu.cn (J.T.); zhuwenying917@163.com (W.Z.); 2Hubei Provincial Clinical Research Center for Precision Radiology & Interventional Medicine, Wuhan 430022, China; 3Hubei Key Laboratory of Molecular Imaging, Wuhan 430022, China; 4Philips Healthcare, Beijing 100600, China; peng.sun@philips.com

**Keywords:** time-dependent diffusion, oscillating gradient, microstructure imaging, breast tumors, MRI

## Abstract

**Objectives**: To explore the feasibility of time-dependent diffusion in clinical applications of breast MRI, as well as the capacity of quantitative microstructural mapping for characterizing the cellular properties in malignant and benign breast tumors. **Methods**: 38 patients with 45 lesions were enrolled. Diffusion MRI acquisition was conducted with a combination of pulsed gradient spin-echo sequences (PGSE) and oscillating gradient spin-echo (OGSE) on a 3T MRI scanner. The microstructural parameters including cellularity extracellular diffusivity (D_ex_), mean cell size, intracellular volume fraction (ν_in_), and the apparent diffusion coefficient (ADC) values were calculated. Each parameter was compared using the unpaired *t*-test between malignant and benign tumors. The area under the receiver operating characteristic curve (AUC) values was used to evaluate the diagnostic performance of different indices. **Results**: The mean diameter, D_ex_, ADC_0Hz_, ADC_25Hz_, and ADC_50Hz_ were significantly lower in the malignant group than in the benign group (*p* < 0.001), while ν_in_ and cellularity were significantly higher in the malignant group (*p* < 0.001). All the microstructural parameters and time-dependent ADC values achieved high accuracy in differentiating between malignant and benign tumors of the breast. For microstructural parameters, the AUC of the cellularity was greater than others (AUC = 0.936). In an immunohistochemical subgroup comparison, the PR-positive group had significantly lower ν_in_ and cellularity, and significantly elevated D_ex_ and ADC_0Hz_ compared to the negative groups (*p* < 0.05). When combining diffusion parameters (cellularity, diameter, and ADC_25Hz_), the highest diagnostic performance was obtained with an AUC of 0.969. **Conclusions**: DWI with a short diffusion time is capable of providing additional microstructural parameters in differentiating between benign and malignant breast tumors. The time-dependent diffusion MRI parameters have the potential to serve as a non-invasive tool to probe the differences in the internal structures of breast lesions.

## 1. Introduction

Breast cancer is one of the most common cancers among women globally [1]. Magnetic resonance imaging (MRI) possesses distinct advantages in the diagnosis and prognostic evaluation of breast lesions. In the current clinical practice, the cornerstone of MRI protocols remains dynamic T1-weighted contrast-enhanced sequences. However, the use of gadolinium-based contrast agents poses health risks including immediate adverse effects, gadolinium deposition, and the development of nephrogenic systemic fibrosis [2,3,4]. Therefore, multiparametric MRI such as diffusion-weighted imaging plays an indispensable role in the diagnosis and evaluation of breast diseases [5].

Diffusion-weighted imaging (DWI) has been investigated as a means of displaying the microstructure of breast lesions and is used to detect changes in the apparent diffusion coefficient (ADC) to reflect the cellular nature of the tumor [6,7,8,9]. However, ADC only reflects the overall measurement of the water diffusion rate and is determined by a variety of microstructure characteristics, such as intracellular and extracellular space, cell size, permeability, and inherent diffusion rate [10,11,12]. Therefore, more accurate and specific imaging markers are needed to display specific tumor microstructures to accurately describe the pathological characteristics of breast tumors.

In current clinical settings, conventional DWI is predominantly implemented through the pulsed gradient spin-echo (PGSE) sequences [13,14], which typically apply diffusion times in the tens of milliseconds. However, the detection sensitivity of DWI is related to diffusion time, and the long diffusion gradient time of PGSE sequences limits the minimum microstructure size that can be measured [15].

Several studies have suggested that the oscillating gradient spin-echo (OGSE) sequence is accessible for microstructural detection [16,17,18]. The oscillatory gradient spin-echo acquisition improves detection sensitivity by shortening the effective diffusion time, and detects the time-dependence of the apparent diffusion coefficient (ADC) through a mixture of long and short diffusion times [17]. Therefore, OGSE sequences may detect the microscopic internal structure of pathological lesions by analyzing the diffusion rate changes of different diffusion times, which overcomes the shortcomings of conventional PGSE sequences [19].

Recently, DWI with OGSE sequences has been employed in animal and clinical human studies of breast cancer [20,21], head and neck tumors [22], glioblastoma [23,24,25], uterine endometrial cancer [26], and prostate cancer [27]. In mouse models of glioma and breast cancer, this approach has demonstrated that the transient diffusion is applicable in cell size measuring and tumor microstructure evaluation [20,25,28]. This kind of time-dependent diffusion MRI is thought to provide more detailed information about the microstructure of tissues. Although OGSE sequences have been used to assess cellular sizes in both cell experiments in vitro and human experiments in vivo of breast cancer, the feasibility of microstructural mapping in patients with benign breast tumors and the diagnostic value of time-dependent diffusion parameters between malignant and benign tumors in clinical MRI systems need to be further explored.

Thus, this work aimed to investigate the feasibility of the time-dependent diffusion using OGSE sequence in clinical applications of breast MRI, the capability of derived microstructural parameters for characterizing cellular features of breast tumors, and to evaluate the accuracy of the microstructural parameters in discriminating between malignant and benign breast disease.

## 2. Materials and Methods

### 2.1. Patient

The prospective study was approved by the institutional ethics committee. The patients with Breast Imaging Reporting and Data System (BI-RADS) category 4 and above lesions detected by ultrasound were enrolled in this study. From October 2023 to May 2024, 69 patients who matched our enrollment criteria underwent breast MRI examination subsequent to perusing the informed consent. The exclusion criteria were as follows: (a) patients without obvious lesions or with a mass diameter smaller than 5 mm (n = 8); (b) lack of biopsy or surgical pathological data (n = 13); or (c) insufficient image quality caused by prominent motion artifacts or low SNR (n = 10). A total of 38 patients with 45 lesions (18 benign, 27 malignant) were ultimately included in the final analysis.

### 2.2. MRI Data Acquisition

All examinations were executed on a 3T MRI scanner using a dedicated 16-channel phased-array breast coil (Ingenia CX, Philips Healthcare, Best, The Netherlands), the patient was placed in a prone position with head first during the entire scanning process. For capturing the diffusion time dependency of various microstructure components in different breast tumors, the time-dependent DWI acquisition was conducted utilizing a combination of two distinct sequences: the oscillating gradient spin-echo (OGSE) and the pulsed gradient spin-echo (PGSE) sequences (Figure 1). The details of acquisition sequence parameters were TR/TE = 4000/105 ms; FOV = 192 × 192 mm; reconstructed in-plane resolution = 2 × 2 mm; slice thickness = 5 mm. The OGSE data were acquired at oscillating frequencies of 25 Hz (effective diffusion time = 10 ms; one cycle; b = 0, 250, 500, 750, and 1000 s/mm^2^; number of averages for each b-values = 1, 1, 2, 3, 4; acquisition time 4 min 12 s) and 50 Hz (effective diffusion time = 5 ms; two cycles; b = 0, 100, 200, and 250 s/mm^2^; number of averages for each b-values = 1, 1, 1, 1; acquisition time 2 min 8 s), and pulsed gradient spin-echo at effective diffusion time of 78.4 ms (b = 0, 250, 500, 750, 1000, 1400, and 1800 s/mm^2^; number of averages for each b-values = 1, 1, 2, 3, 4, 5, 6; acquisition time 4 min 24 s). The routine MR imaging examination was based on the subsequent protocol: T2 short-time inversion recovery (TR 4074 ms, TE 65 ms, thickness 3.3 mm, FOV 280 × 339 mm). For dynamic contrast-enhanced imaging, one pre-contrast and five post-contrast consecutive scans were performed using 3D T1 high-resolution isotropic volume sequence with fat suppression (TR 4.0 ms, TE 2.1 ms, section thickness 1.5 mm, FOV 200 × 357 mm). Gadobenate (Multihance, BRACCO, Milano, Italy) was selected as the contrast agent, injected at a dose of 0.1 mmol/kg via a high-pressure injector, and subsequently flushed with 15 mL saline.

### 2.3. Image Analysis

Image pre-processing was performed as follows: first, the original PGSE and OGSE sequence generated DICOM image was transformed into NIFTI format using the MRtrix 3.0 software (https://www.mrtrix.org/) (accessed on 13 December 2023); the original PGSE and OGSE images were denoised using DWI denoising module; the Gibbs–Ring artifact removal was carried out by mrdegibbs module, and then the image was rigidly aligned using the mcflirt module of FSL version 6.0.3 software, as reported previously [29,30]. After necessary preprocessing, the diffusion parametric maps were fitted using in-house written MATLAB scripts consulting the open-source code of the imaging microstructural parameters using a limited spectrally edited diffusion (IMPULSED) scheme (https://github.com/jzxu0622/mati.git) (accessed on 27 December 2023). According to the analytical expressions of the OGSE and PGSE sequence parameters reported in the previous studies [18,31,32,33], the following parameters are obtained: cell size (d), extracellular diffusivity (D_ex_), and intracellular volume fraction (ν_in_). Cellularity was represented as ν_in_/d × 100 in this study [27]. The physiologically relevant values served as the basis for setting the range of fitted parameters: 5 ≤ d ≤ 50 μm, 0 ≤ D_ex_ ≤ 3.0 μm^2^/ms, and 0 ≤ ν_in_ ≤ 1.

Two experienced radiologists (S.P. and J.L., with 6 and 15 years of experience, respectively) manually defined a region of interest (ROI) at the maximal level of the lesion on PGSE diffusion-weighted images (with b = 1800 s/mm^2^), and copied them on the corresponding OGSE diffusion-weighted images and other derived parametric maps. The ROI masks were then eroded to reduce the edges of the mask to avoid partial volume effects and artifacts near the margin. The mean values of each IMPULSED-derived parameter (d, ν_in_, and D_ex_), the ADC values for each OGSE (ADC_25Hz_ and ADC_50Hz_) sequence, and PGSE sequences (ADC_0Hz_) were calculated for the entire ROI.

### 2.4. Histopathologic Analysis

The pathologist verified the ultimate pathologic diagnosis based on surgical excision (n = 36) or biopsy (n = 9) specimens. Further immunohistological analysis was performed on 27 malignant lesions. According to estrogen receptor (ER) status, progesterone receptor (PR) status, human epidermal growth factor receptor-2 (HER-2) status, and Ki67 index, tumors were divided into four subgroups: luminal A, luminal B, HER2-positive, and triple-negative [34].

### 2.5. Statistical

Data analysis was performed using MedCalc software (version 22.030, MedCalc Software, Ostend, Belgium). Inter-reader agreement on all quantitative measurements between the two observers was assessed by calculating intraclass correlation coefficients (ICC). The fitted microstructural parameters and ADC values at different diffusion times measured by both observers were averaged for further analysis. The mean values of each IMPULSED-derived parameter (diameter, cellularity, ν_in_, and D_ex_) and ADC values (ADC_0Hz_, ADC_25Hz_, and ADC_50Hz_) between were compared using the unpaired-t test. The diagnostic performance of each parameter for benign and malignant breast tumors was assessed using receiver operating characteristic (ROC) analysis. For malignant tumors, the microstructural parameters in different immunohistological subtype groups were further compared using the unpaired *t*-test. A *p* value of <0.05 was considered statistically significant.

## 3. Results

### 3.1. Patient Characteristics

A total of 38 patients (45.26 ± 10.97 years old) with 45 lesions (27 of malignant, 18 of benign) were successfully enrolled in this study. All the malignant lesions were confirmed as invasive ductal carcinomas (IDC), while benign lesions were composed of 15 fibroadenomas and 3 phyllodes tumors. The characteristics of the tumors are summarized in Table 1.

### 3.2. Microstructural Parameters of Malignant and Benign Breast Tumors

The DW signals were fitted to the IMPULSED signal model to generate cellular parameters, the IMPULSED-derived maps of malignant and benign lesions were shown in Figure 2 and Figure 3. The fitted overall average cell size d, intracellular volume fraction (ν_in_), extracellular diffusivity (D_ex_), cellularity, and different ADC values were compared between malignant and benign breast tumors (Table 2). There were significant differences in all the time-dependent diffusion MRI-derived parameters across benign and malignant breast lesions (*p* < 0.001).

### 3.3. Microstructural Parameters of Different Immunohistochemical Groups in Breast Tumor

The microstructural parameters were compared between ER, PR, HER2, and Ki67 of the positive and negative groups (Figure 4). It showed that ν_in_ was markedly increased in the PR-negative (*p* = 0.012) and Ki67-positive (*p* = 0.014) groups. Cellularity was significantly higher in the PR-negative (*p* = 0.020) group compared to the positive group. D_ex_ was significantly higher in ER-positive (*p* = 0.011), PR-positive (*p* = 0.037), and Ki67-negative (*p* = 0.020) groups. ADC_0Hz_ increased in the ER-positive (*p* = 0.010), PR-positive (*p* = 0.002), and Ki67-negative (*p* = 0.045) groups, and ADC_25Hz_ was significantly higher in the PR-positive (*p* = 0.020) group compared to the negative group (*p* = 0.041).

### 3.4. Diagnostic Performance of Microstructural Parameters

The results of receiver operating characteristic (ROC) curve analyses of the IMPULSED-derived parameters and different ADC values are shown in Table 3 and Figure 5A,B. Pairwise comparisons of the AUCs among the d, ν_in_, cellularity, and Dex revealed that the AUC of the cellularity was significantly greater than that of the ν_in_ (*p* = 0.028), whereas no other comparison of the AUCs revealed significant differences. For different ADC values, ADC_25Hz_ showed the highest performance (AUC = 0.953); however, no significant difference was observed between benign and malignant breast lesions in any of the five indices of ADC_0Hz_, ADC_25Hz_, and ADC_50Hz_.

We next sought to construct models involving different imaging variable combinations to improve prediction performance (Figure 5C). Model 1 (diameter+ cellularity + ν_in_ + D_ex_) combined all the microstructural parameters and yielded an AUC of 0.959 (95%CI: 0.853–0.996). Model 2 (cellularity + diameter + ADC_25Hz_) combined two microstructural parameters and time-dependent ADC values and showed a superior predictive performance (AUC, 0.969 [95%CI: 0.869–0.998]) for malignant breast lesions compared with Model 1 and other single-parameter models.

## 4. Discussion

In this study, we explored the feasibility of time-dependent diffuse MRI for assessing the microstructural features of breast tumors. By using microstructural parameters for the identification of benign from malignant breast tumors, we found that the malignant breast tumor group was characterized by a higher volume fraction, smaller mean cell size, higher cellularity, and lower diffusivities. These IMPULSED-derived parameters achieved excellent accuracy in distinguishing malignant from benign breast tumors, with the highest AUC of 0.936 in cellularity. The result indicated that the OGSE sequence bears the potential to assess the microstructural characteristics of breast tumors, and the OGSE-derived parameters can function as non-invasive biomarkers to probe tumor pathologies of malignant and benign tumors.

Non-invasive breast cancer diagnosis and biomarker identification related to the internal microstructure of tumors were very important in guiding clinical management and prognosis prediction. The traditional PGSE sequences only reflect the internal characteristics of breast tumors from the value of the ADC, and the detection sensitivity of the microstructure in tumor cells is limited by its long diffusion time [14]. We confirmed that OGSE sequences can derive microstructural parameters such as cell diameter, cellular diffusivity, and cell density by a combination of different diffusion times, which overcomes the shortcomings of conventional PGSE sequences. Several studies [20,35] have also verified a significant correlation in mean cell size between the values obtained from light microscopy and IMPULSED fitted results, and the IMPULSED-derived cell size in our study was also similar to that in previous studies. This indicates that the fitted microstructure parameters in our study are reliable for the structural analysis of breast tumors. Those differences may be associated with the high cellularity and heterogeneous microenvironment of breast cancer. Malignant breast tumors were composed of densely arranged cells with high proliferative capacity. As the cellularity increased, both the extracellular space and the diffusion of water were reduced [36,37,38]. Conversely, the benign lesions are characterized by sparsely distributed cells, larger extracellular space, and the less restricted diffusion of water, which leads to higher ADC values [35,39,40]. Moreover, our study evaluated the microstructure parameters of benign tumors, and assessed the differential diagnostic performance of time-dependent diffusion parameters in clinical routines.

It has been demonstrated that the ADC value of a tumor varies with the diffusion time, typically showing a gradual increase as the diffusion time shortens [22,23,24,26]. Regarding the ADC values at different effective diffusion times in the OGSE and PGSE sequences, our results showed that the highest ADC values for the breast lesions were observed in the OGSE_50Hz_ sequence with a diffusion time of 5 ms, while the lowest values in the PGSE sequence with a diffusion time of 78.4 ms. The values of ADC_0Hz_, ADC_25Hz_, and ADC_50H_ in the malignant group were significantly lower than those in the benign group, and demonstrated excellent diagnostic efficacy. Although the diagnostic performance of ADC_25Hz_ seemed a bit better than that of ADC_0Hz_ and ADC_50Hz_, no significant difference was detected. Recently, a number of studies have shown interest in the ADC values associated with time-dependent diffusion MRI for the differential diagnosis of tumors and prognosis prediction [41,42]. Iima et al. [22] reported that malignant head and neck tumors exhibited lower ADC_OGSE_ and ADC_PGSE_ values, and the relative ADC changes were significantly larger in malignant groups. Kamimura et al. [23] examined patients with glioblastoma and primary lymphoma and found that the time-dependent ADC values showed excellent diagnostic performance in differentiation diagnosis. Ejima et al. [26] indicated that the ADC_OGSE_/ADC_PGSE_ could be used for the histological grading and prognostic risk prediction of uterine endometrial cancer. However, there is insufficient evidence to show that the diagnostic performance of ADC_OGSE_ is significantly superior to that of ADC_PGS_.

In an immunohistochemical subgroup comparison, our results indicated that the breast cancers with positive PR had significantly lower ν_in_, cellularity, elevated D_ex_, and ADC_0Hz_ compared to the negative groups. This result agrees with the fact that positive ER or PR status commonly pointed to a low-grade breast tumor with a favorable prognosis, and was associated with lower cellularity, vascularity, and aggressiveness [43,44,45]. Moreover, we also found that breast cancers with high Ki67 expression had higher ν_in_ and lower D_ex_, which may relate to the high proliferation and adverse prognosis of the tumor [43]. Ba et al. [21] reported that the PR-positive groups demonstrated reduced cellularity and increased diffusivities, while the HER2-negative groups presented with smaller cell diameters. These findings could be associated with a higher degree of malignancy and adverse prognosis in tumors with HER2 overexpression. However, our study found no significant differences in microstructural parameters between the HER2-positive and -negative groups. Although time-dependent diffusion has shown potential in distinguishing the immunophenotypes, the accuracy and diagnostic performance of microstructural parameters derived by time-dependent diffusion in discriminating immunophenotypes in breast tumors demand further exploration.

This study has several limitations. First, the study was conducted on a single MR system at a single center. MRI systems with different magnetic field strengths (1.5 T, 3 T, or even higher field strengths) exhibit differences in characteristics such as the uniformity of the magnetic field. Therefore, the parameters of the OGSE sequence need to be adjusted according to the characteristics of the equipment and the imaging requirements, and the reproducibility of this technology in different MRI systems still needs to be further verified. Second, the relatively small sample size may result in sample selection bias; therefore, our suggested threshold microstructure parameters (diameter, cellularity, ν_in_, and D_ex_) might not be representative of those of a larger population, and a dedicated evaluation is needed before OGSE can be truly useful in clinical practice. Third, histopathological validation was not performed, and it is imperative to conduct further examination to clarify the correlation between the IMPULSED-fitted microstructural and histopathological-based measurements.

## 5. Conclusions

Our study indicated that the time-dependent diffusion MRI bears the potential to evaluate the microstructural characteristics of breast lesions, and may be useful in the differential diagnosis of benign and malignant breast lesions. However, our research findings need to be further validated in larger multicenter cohorts and applied for clinical examination.

## Figures and Tables

**Figure 1 diagnostics-15-00823-f001:**
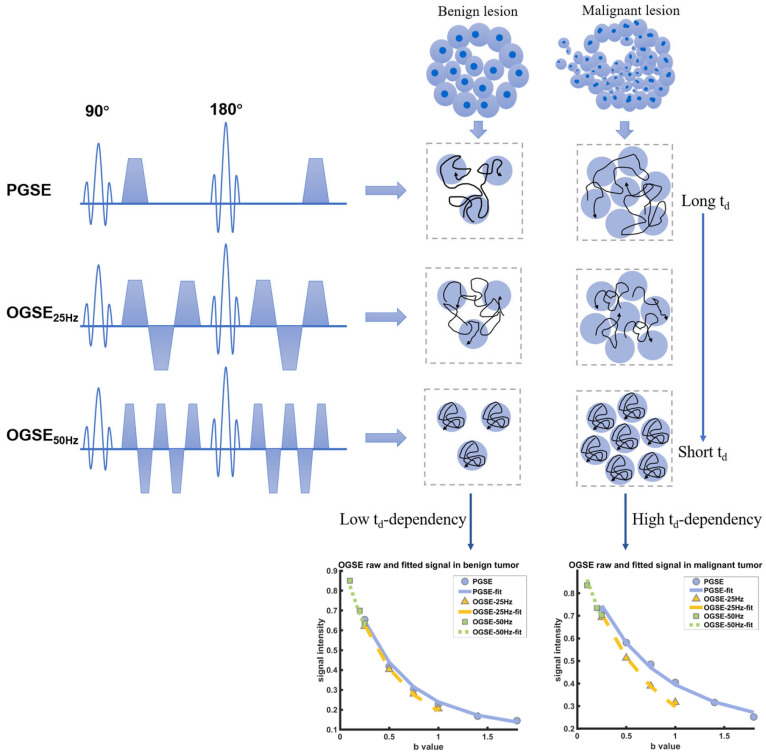
Schematic shows the pulse sequences used in the IMPULSED data acquisitions, in addition to conventional PGSE acquisitions, OGSE acquisitions with two frequencies (N = 25 Hz and 50 Hz) are used. The diffusivity of water molecules measured in a cellular environment is dependent on diffusion time, and the diffusion time dependence increases with cellularity increasing. The fitted results of the DW signals in varying diffusion times indicate the higher diffusion time dependence in malignant breast tumors.

**Figure 2 diagnostics-15-00823-f002:**
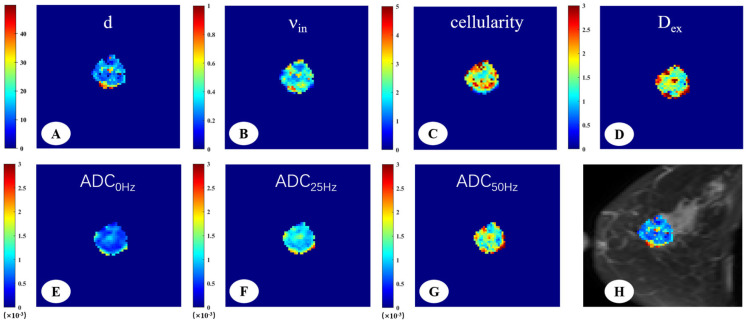
A 66-year-old woman with invasive ductal carcinoma in the left breast. The IMPULSED-derived maps showed mean cell size d = 16.47 μm (**A**), intracellular volume fraction ν_in_ = 0.41 (**B**), cellularity = 2.86 (**C**), and extracellular diffusion coefficient Dex = 1.82 μm^2^/ms (**D**). The ADC map derived from PGSE (**E**), OGSE_25Hz_ (**F**), and OGSE_50Hz_ (**G**) showed the ADC values in different sequences, ADC_0Hz_ = 0.72 × 10^−3^ mm^2^/s, ADC_25Hz_ = 1.18 × 10^−3^ mm^2^/s, ADC_50Hz_ = 1.65 × 10^−3^ mm^2^/s. IMPULSED-derived maps of mean cell size d overlaid on the corresponding diffusion-weighted image (**H**).

**Figure 3 diagnostics-15-00823-f003:**
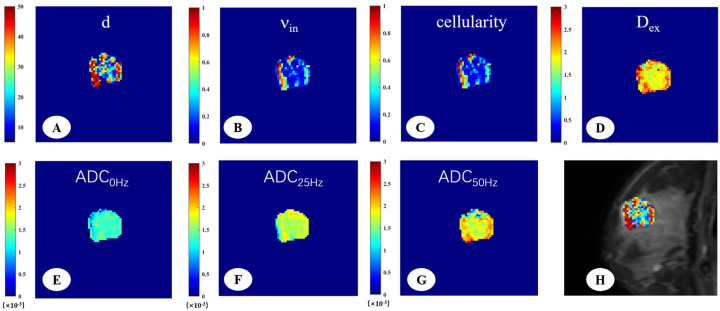
A 33-year-old woman with fibroadenoma in the right breast. The IMPULSED-derived maps showed mean cell size d = 29.55 μm (**A**), intracellular volume fraction ν_in_ = 0.31 (**B**), cellularity = 1.01 (**C**), and extracellular diffusion coefficient D_ex_ = 2.00 μm^2^/ms (**D**). The ADC map derived from PGSE (**E**), OGSE_25Hz_ (**F**), and OGSE_50Hz_ (**G**) showed the ADC values in different sequences, ADC_0Hz_ = 1.25 × 10^−3^ mm^2^/s, ADC_25Hz_ = 1.68 × 10^−3^ mm^2^/s, ADC_50Hz_ = 1.84 × 10^−3^ mm^2^/s. IMPULSED-derived maps of mean cell size d overlaid on the corresponding diffusion-weighted image (**H**).

**Figure 4 diagnostics-15-00823-f004:**
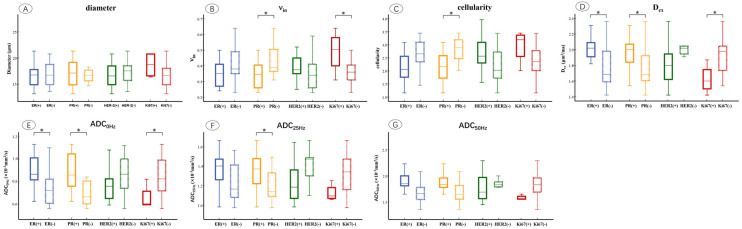
Box and whisker plots show the comparisons of microstructural parameters between different ER, PR, HER2, and Ki67 groups, including mean diameters (**A**), ν_in_ (**B**), cellularity (**C**), D_ex_ (**D**), ADC_0Hz_ (**E**), ADC_25Hz_ (**F**), ADC_50Hz_ (**G**). *p* < 0.05 was denoted in the plots using *. Whiskers denote the range in each group, dots represent individual data points, boxes indicate the SD, and midlines are the median.

**Figure 5 diagnostics-15-00823-f005:**
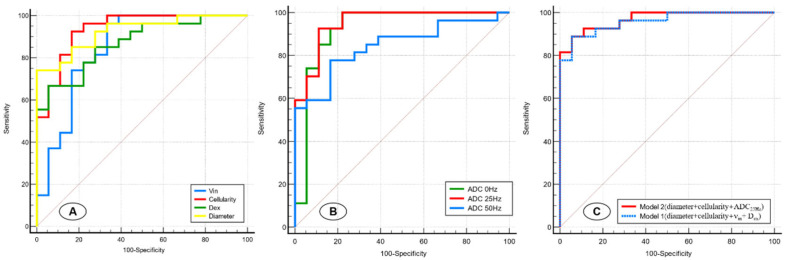
Receiver operating characteristic curves for the IMPULSED-derived parameters (d, ν_in_, cellularity, and D_ex_) (**A**), different ADC values (ADC_0Hz_, ADC_25Hz_, and ADC_50Hz_) (**B**), and two models involving different variable combinations (**C**).

**Table 1 diagnostics-15-00823-t001:** The characteristics of all lesions.

Characteristics	Number
Ages (years)	45.26 ± 10.97
Benign	
Fibroadenomas	15 (83.3%)
Phyllodes	3 (16.7%)
Malignant	
Invasive ductal carcinoma	27 (100%)
ER status	
Positive	14 (51.9%)
Negative	13 (48.1%)
PR status	
Positive	11 (40.7%)
Negative	16 (59.3%)
HER2 status	
Positive	11 (40.7%)
Negative	16 (59.3%)
Ki67	
<14%	4 (14.8%)
≥14%	23 (85.2%)
Tumor subtype	
Luminal A	4 (14.8%)
Luminal B	10 (37.0%)
HER2-positive	7 (25.9%)
Triple-negative	6 (22.2%)

Estrogen receptor (ER), progesterone receptor (PR), human epidermal growth factor receptor type 2 (HER2).

**Table 2 diagnostics-15-00823-t002:** Group comparison between the estimated microstructural parameters.

	Malignant	Benign	*p* Value
n_in_	0.38 ± 0.10	0.24 ± 0.10	<0.001 *
d (μm)	17.26 ± 2.88	24.13 ± 4.54	<0.001 *
Cellularity	2.48 ± 0.68	1.13 ± 0.53	<0.001 *
D_ex_ (μm^2^/ms)	1.88 ± 0.25	2.24 ± 0.17	<0.001 *
ADC_0Hz_ (×10^−3^ mm^2^/s)	0.88 ± 0.12	1.24 ± 0.21	<0.001 *
ADC_25Hz_ (×10^−3^ mm^2^/s)	1.30 ± 0.19	1.77 ± 0.18	<0.001 *
ADC_50Hz_ (×10^−3^ mm^2^/s)	1.81 ± 0.28	2.16 ± 0.24	<0.001 *

* indicate that the *p*-values are statistically significant (*p* < 0.05).

**Table 3 diagnostics-15-00823-t003:** Diagnostic performance of IMPULSED-derived microstructural parameters and different ADC values in differentiating malignant (n = 27) and benign (n = 18) tumors.

Parameter	AUC (95%CI)	Sensitivity	Specificity
Microstructural parameters			
n_in_	0.848 (0.7009–0.937)	0.963	0.667
d (μm)	0.926 (0.807–0.983)	1.000	0.741
Cellularity	0.936 (0.821–0.987)	0.926	0.833
D_ex_ (μm^2^/ms)	0.870 (0.737–0.952)	0.944	0.667
ADC values			
ADC_0Hz_ (×10^−3^ mm^2^/s)	0.924 (0.805–0.982)	0.778	1.000
ADC_25Hz_ (×10^−3^ mm^2^/s)	0.953 (0.844–0.944)	0.926	0.889
ADC_50Hz_ (×10^−3^ mm^2^/s)	0.846 (0.707–0.936)	0.778	0.833
Model 1	0.959 (0.853–0.996)	0.889	0.944
Model 2	0.969 (0.869–0.998)	0.889	0.944

## Data Availability

The datasets generated during and/or analyzed during the current study are available from the corresponding author on reasonable request.

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
