# Peer review of "Imaging Microstructural Parameters of Breast Tumor in Patient Using Time-Dependent Diffusion: A Feasibility Study"

_diagnostics, 2025, doi:10.3390/diagnostics15070823_

Round 1

Reviewer 1 Report

Comments and Suggestions for Authors

Dear Authors,

Thank you very much for allowing me to express my opinions related to your work. As a researcher myself, I admire and respect the effort you put into constructing your study and building this manuscript.

Bellow, you can find my comments regarding certain issues. I hope these comments will help you improve both your current and future work.

Abstract

  • The abstract clearly presents the study’s objectives, methods, results, and conclusions. However, it could benefit from a clearer statement about the clinical implications of using time-dependent diffusion MRI in breast tumor characterization.
  • Consider briefly defining key terms such as "cellularity" and "Dex" for clarity to a broader audience.
  • The conclusion could be more specific about how this method improves upon conventional techniques.

Introduction

  • The introduction provides a strong rationale for the study, explaining the limitations of conventional diffusion MRI and the potential of OGSE-based imaging.
  • Some references to prior studies could be better integrated to highlight the novelty of this work.
  • The last paragraph effectively sets up the research question, but the transition to the Materials and Methods section could be smoother.

Materials and Methods

  • The patient selection criteria are well defined, but the justification for the exclusion criteria could be expanded.
  • The MRI acquisition parameters are described in detail, but it might be helpful to include a brief rationale for the specific choice of OGSE frequencies and diffusion times.
  • The statistical analysis is thorough, but it would be useful to mention how missing or poor-quality data were handled.

Results

  • The presentation of results is clear and well-structured, with appropriate use of tables and figures.
  • The ROC analysis is particularly valuable, but additional interpretation of how the AUC values compare to previous studies could strengthen the section.
  • Some of the subgroup analyses (e.g., immunohistochemical markers) would benefit from more discussion on their clinical significance.

Discussion

  • You should start the Discussion section by adressing your results. In this particular case, you can move up the second paragraph and make some slight adjustments.
  • The discussion effectively interprets the findings and compares them to prior studies.
  • The limitations section is appropriate but could include more discussion on the reproducibility of the technique across different MRI systems.
  • As a limitation to your study you should also mention that due to the retrospective design of this study, there may have been selection bias.
  • Are there any limitations provided by the software used for data analysis?
  • I would suggest also addressing new and emerging techniques used in breast MRI for assessing tumor histology but also treatment response and prognosis (such as texture analysis), and in particular studies that evaluate the role of DWI-based texture analysis that uses MRI images for breast cancer evaluation

Conclusion

  • The conclusion summarizes the findings well but could be more explicit about the next steps for clinical implementation.
  • A brief mention of potential future research directions (e.g., validation in larger, multicenter studies) would be useful.

Author Response

Dear Reviewer,
Thank you so much for the time and effort you've invested in evaluating our submission. Your work has been invaluable in helping us identify areas for improvement. To answer your comments, we have prepared an attached document that details our responses and the corresponding revisions made to the manuscript. We sincerely hope this addresses all the issues you raised.

Reviewer 2 Report

Comments and Suggestions for Authors

The authors evaluated 45 lesions in 38 breast patients with MRI according to their cellular characteristics in the differentiation of benign and malignant lesions.

Recommendations
1. In the introduction, it is stated that MRI has important advantages in the prognostic evaluation of breast lesions. There is no proven place for MRI in the prognostic evaluation of breast lesions with prospective randomized studies.

2. Is this a retrospective study? This should be explained.

3. The number of patients is quite insufficient for such a study and the lack of histopathological confirmation weakened the study.

Author Response

(The authors gave the same response as above.)

Reviewer 3 Report

Comments and Suggestions for Authors

In this study the authors tested time-dependent diffusion for teh diagnosis of breast cancer. They showed that IMPULSED parameters were differnt for breast cancer and benign lesions. Cancers has higher intracellular fraction and cellularity and lower cell diameter, diffusivity and ADC. There are a few concerns that authors need to address prior to publication.  

Comments

Title  - Temporal Diffusion Spectroscopy  only appears in the conclusion (line 378). I suggest using "time-dependent diffusion" as that is what the methods is descried in most of the text.

Abstract - OK

Introduction - line 69-70 clearly mention that time-dependent diffusion has been tested in the breast in studies referenced 20 and 21. These studies included in-vivo human clinical scan on 3T scanners and included OGSE sequences. Based on this, the novelty of this study  that the authors claim in lines 75-78 doesnt hold up. So, the authors needs to clearly state what is new about their study. This is my major concern.

Methods-

  • line 87- Clearly state that  it was a prospective study.
  • Add the scan time for each of the sequences, number of averages for each b-values.
  • Was any fat saturation  used in DWI?

Results - 

  • Fig 2 and 3 - ROIs in the corresponding DWI images (H), seem smaller than the ones seen in parameter maps. Why is that? Why did they not show the parameter map for the whole breast and outline the ROI? This way readers would appreciate the difference in IMPULSE parameters in the cancer  and benign lesion versus the surrounding benign tissue.
  • table 4 - Report at what cutoff point was the sensitivity and specificity reported for. Was it the Youndens index?

Disussion - OK

Author Response

(The authors gave the same response as above.)

Round 2

Reviewer 2 Report

Comments and Suggestions for Authors

The authors responded to criticisms and comments in good faith. However, the information provided at the end of the article is very important and significant for science. It is quite restrictive to state such an important message with the numbers of 18 benign and 27 malignant breast patients.

Author Response

Dear Reviewer,

We sincerely appreciate your recognition of our response. We fully concur with your perspective regarding the sample sizes and the conclusion of this study.

To answer your comments, we have prepared an attached document that details our responses and the corresponding revisions made to the manuscript. We sincerely hope this addresses all the issues you raised.

Best regards,

Shuyi Peng
